# The Germination Performance After Dormancy Breaking of *Leucaena diversifolia* (Schltdl.) Benth. Seeds in a Thermal Gradient and Its Distribution Under Climate Change Scenarios

**DOI:** 10.3390/plants13202926

**Published:** 2024-10-18

**Authors:** Andrés Flores, Cesar M. Flores-Ortíz, Patricia D. Dávila-Aranda, Norma Isela Rodríguez-Arévalo, Salvador Sampayo-Maldonado, Daniel Cabrera-Santos, Maraeva Gianella, Tiziana Ulian

**Affiliations:** 1CENID-COMEF, National Institute for Forestry, Agriculture and Livestock Research, Progreso 5, Coyoacán 04010, Mexico; flores.andres@inifap.gob.mx; 2National Laboratory in Health, FES Iztacala, UNAM, Tlalnepantla 54090, Estado de Mexico, Mexico; 3Plant Physiology Laboratory, UBIPRO, FES Iztacala, UNAM, Tlalnepantla 54090, Estado de Mexico, Mexico; ssampayom@hotmail.com (S.S.-M.);; 4Natural Resources, UBIPRO, FES Iztacala, UNAM, Tlalnepantla 54090, Estado de Mexico, Mexico; pdavilaa@unam.mx (P.D.D.-A.); isela.unam@gmail.com (N.I.R.-A.); 5Royal Botanic Gardens, Kew Wellcome Trust Millennium Building, Wakehurst, Ardingly, West Sussex RH17 6TN, UK; m.gianella@kew.org (M.G.); t.ulian@kew.org (T.U.); 6University of Turin, 10147 Turin, Italy

**Keywords:** cardinal temperatures, climate change, Fabaceae, legume, seed testing, thermal time

## Abstract

Climate change models predict temperature increases, which may affect germination, an important stage in the recruitment of individuals in agroecosystems. Therefore, it is crucial to conduct research on how temperature will impact the germination of multipurpose native species. *Leucaena diversifolia* (Schltdl.) Benth. is native to America and is commonly cultivated around the world due to having a high protein content in seeds, and their trees are used in agrosilvopastoral systems because they fix nitrogen and provide shade and cattle feed. However, climate change affects the critical phases of its life cycle and influences its growth, reproduction, phenology, and distribution. To assess the germination performance of *Leucaena diversifolia* under different temperatures throughout thermal times, we estimated germination variables and determined cardinal temperatures and thermal time; we also analysed germination and potential distribution under two climate change scenarios. We found significant variations in seed germination (78–98%) and differences in cardinal temperatures (*Tb* = 5.17 and 7.6 °C, *To* = 29.42 and 29.54 °C, and *Tc* = 39.45 and 39.76 °C). On the other hand, the sub-optimal and supra-optimal temperature values showed little differences: 51.34 and 55.57 °Cd. The models used showed variations in germination time for the analysed scenarios and the potential distribution. We confirm that the populations and distribution of *L. diversifolia* will be altered due to climate changes, but the species retains the ability to germinate under warmer conditions.

## 1. Introduction

In recent decades, research efforts focused on proposing species for agroforestry systems with high potential to meet the demand for forest products and environmental benefits in rural areas, such as fruits, wood, firewood, seeds, cattle food, soil fertility improvement, nitrogen fixation, carbon sequestration, temperature and wind moderation, and biodiversity enhancement [1,2]. These works align with preserving landscapes, agrobiodiversity, and traditional knowledge [3]. For some species, there has been extensive research, e.g., legume species (*Leucaena* spp.) [4], mahogany (*Swietenia macrophylla* King) [5], and cedar (*Cedrela odorata* L.) [6], but most of the tree species that reside in the wet climate are still unexplored.

*Leucaena diversifolia* (Schltdl.) Benth. is one of the 22 species reported for the genus *Leucaena* in America [7] and is widely studied and utilised [8]. Most of its distribution is located in southwestern Mexico in the state of Oaxaca [9], established in different conditions (soils and climatic zones) in sub-humid and humid areas, with an altitude range of 300–2000 m [10], and it grows in cold climates [9,11] due to its tolerance of frost [10]. This species flowers between December and February [10], its seeds present physical dormancy due to the impermeability of the seed coat to water [12], and is cultivated around the world due to having a high level of protein content in seeds, and their trees are used in agrosilvopastoral systems because they fix nitrogen, and provide shade and cattle feed [9]. The trees are used as firewood, and for building, recovering degraded lands, and shade for crops (coffee and cocoa) [9]. These characteristics make this species very important in Mexico, particularly for agroforestry systems.

Studies of the effect of temperature on germination are important, since germination is an important stage for the recruitment of individuals to plant communities and can be significantly affected by changes in temperature. The phenology of forests is changing due to temperature increases and precipitation decreases brought on by climate change, which may have an impact on crucial life cycle stages (flowering, seed development, and seedling establishment) [13], and influence their growth, reproduction, phenology, and distribution [14]. By 2030, it is predicted that Mexico’s mean annual temperature will rise by 1.5 °C and its yearly precipitation will fall by −6.7% [15], which can change the germination rate in trees. Due to predicted variations in temperature and water supply, germination performance is a fundamental trait that needs to be studied to understand the species’ resilience under different climate change scenarios.

Germination (radicle emergence) is the most critical stage of the plant’s life cycle [16] because it can influence the efficient use of water, which nutrients are available to emerge, and seedling development [17]. For seed germination and species distribution, temperature plays one of the most important roles [18]. Germination speed is an important parameter to understand before establishing any species in the field, as it indicates its ecological advantage under natural conditions [19], helping it adapt to habitat conditions, life-cycle strategy, and local climate. Faster germination can be especially advantageous for species that benefit from early germination, allowing for the rapid occupation of an ecosystem [20]. In order to explore seed germination or other developmental phases (such as flowering or attaining seed physiological maturity) under climate change scenarios, thermal time has been used. There are five parameters of the thermal niche of seeds: the three cardinal temperatures or thermal thresholds (base or minimum (Tb), optimal (To), and ceiling or maximum (Tc)), and the thermal time (sub- and supra-optimal) [21]. These thermal times have been estimated for important species from Mexico [22,23,24,25], but there is no recorded information for *Leucaena diversifolia*. This research aimed to describe the germination performance of *Leucaena diversifolia* under different temperatures using thermal times models. The following questions were posed: (1) Does the temperature range 5–35 ± 2 °C affect seed germination, germination time (T_50_), and germination rate? (2) What are the cardinal temperatures for the target species? (3) Does the potential distribution change under climate change scenarios?

## 2. Results

### 2.1. Germination

Temperature significantly affected the proportion of germinated seeds, irrespective of their range [$X^{2}\left(5,N=30\right)=41.79,\rho <0.0001$]. The average germination over the temperature range accounts for 90%, with the highest germination reaching 98% observed at 20 °C and 25 °C. At 35 °C, 78% of the seeds germinated, which was the lowest value (Figure 1). The mean germination time (T_50_) was also significantly affected by temperature [$X^{2}\left(5,N=30\right)=127.15,\rho <0.0001$]. For 50% germination, seeds required less time at higher temperatures: 2 days at 30 °C, 3 days at 20 °C and 25 °C, and 5 days at 35 °C. Conversely, more time was needed at lower temperatures, i.e., 12 and 18 days at 15 °C and 10 °C, respectively.

The temperature range produced significant differences for the germination rate [$X^{2}\left(5,N=30\right)=140.54,\rho <0.0001$]. The fastest germination occurred at 30 °C, with 10 seeds germinating per day. However, the germination speed considerably decreased at 35 °C, with only four seeds germinating per day. At 10–15 °C, the germination speed was the slowest, with one seed germinating per day. At 20–25 °C, the germination speed was moderate, with four seeds germinating per day at 20 °C and three seeds per day at 25 °C.

### 2.2. Cardinal Temperatures

The average cardinal temperatures estimated by the model with two-segment regression lines (M1) were as follows: 5.17 °C for Tb, 29.42 °C for To, and 39.45 °C for Tc. On the other hand, for the model with simple regression lines (M2), the estimated cardinal temperatures were as follows: 7.6 °C for Tb, 29.54 °C for To, and 39.76 °C for Tc (Figure 2).

For model M1, the predicted *Tc* for 80% was 35 °C, which affected the average value (39.45 °C); this is due to the 35 °C limit for the inflexion point, the temperature at which this percentile ceases germination. The same was observed for 80% in the M2 model; however, the predicted *Tc* for this model (39.76 °C) was higher than the M1 model, where no inflexion points were added to the regression lines (Figure 2, Table 1 and Table 2).

The goodness of fit output parameters showed quite similar tendencies for both models. For model M1 (Table 1), in the sub- and supra-optimal ranges, RMSE tended to decrease from 10% to 70%, but an increase was observed in the RMSE value for 80%. R^2^ and adjusted R^2^, on the other hand, tended to rise from 10% to 70%, while R^2^ values decreased at 80%, indicating a lower fit for this percentile. The same tendencies were observed for model M2 (Table 2).

### 2.3. Thermal Time

For sub-optimal temperatures $\left(\theta_{1}\left(50\right)\right)$, the thermal time to reach 50% germination was 51.34 ± 2.89 °Cd, based on the Probit model, with an R^2^ value of 75.88 (Table 3). In the supra-optimal temperature range $\left(\theta_{2}\left(50\right)\right)$, the thermal time estimate for 50% germination was 55.57 ± 2.85 °Cd, with an R^2^ value of 76.2593 for the Probit model (Table 3).

Figure 3 shows that the probability of achieving a higher germination proportion rises with increasing heat accumulation for thermal time at sub-optimal temperatures. This process approaches a thermal time $\left(\theta_{1}\left(50\right)\right)$ of 51.34 ± 2.89 °Cd, at which 50% of the seed lot germinates. On the other hand, in the case of thermal time within the supra-optimal range, the germination fraction decreases with increasing heat because the thermal time $\left(\theta_{2}\left(50\right)\right)$ was found to be 55.57 ± 2.85 °Cd (Figure 3).

### 2.4. Germination and Potential Distribution Under Climate Change Scenarios

For the target species, seed dispersion occurs in April, with an average temperature of 20.3 °C. Temperature increases were predicted by the French model (CNRMCM5) to be 1.7 °C, the English model (GFDL-CM3) to be 1.9 °C, the German model (MPI-ESM-LR) to be 2.1 °C, and the American model (GFDL-CM3) to be 2.5 °C. These estimates were made under an intermediate future scenario (2050). However, the values changed under a distant-future scenario (2090). The French model showed a temperature increase of 2.0 °C, 2.5 °C with the English model, 2.3 °C with the German model, and 2.9 °C with the American model.

Regarding the thermal time ($\theta_{1}$(50)), the models showed that it accumulates in 3.4 days in the current scenario. However, under an intermediate future scenario (2050), it reached 0.35 and 0.4 days earlier than the current scenario with the French and English models, respectively, and 0.43 and 0.49 days earlier with the German and American models (Figure 4). Conversely, models based on a distant-future and conservative scenario suggest that the thermal time is reached 0.41 days earlier with the French model and 0.46 days earlier with the German model compared to the current scenario, and 0.49 and 0.56 days earlier with the English and American models (Figure 5).

### 2.5. Potential Distribution

The current surface area of *L. diversifolia* (65,971.63 km^2^) is projected to decrease in both medium-term and distant-future scenarios (2050 and 2090). The reduction in surface area is expected to be higher in the distant future compared to the medium-term future, ranging from 26.5% to 12.1%. Under scenarios with higher CO_2_ emissions (8.5 Watts/m^2^), the potential surface area reduction is projected to be higher compared to scenarios with lower CO_2_ emissions (2.6 Watts/m^2^), ranging from 26.5% to 6.5% (Figure 6). In the distant future under the SSP5-8.5 scenario, states like Colima, Tamaulipas, Tlaxcala, and Michoacán are expected to be most affected by climate change, experiencing a reduction in surface area of over 90%. However, increased temperature and precipitation may lead to expansion to new areas in Guerrero, Jalisco, Estado de México, Hidalgo, and Puebla (around 240%). States like Querétaro, Morelos, and San Luis Potosí states are projected to be relatively less affected (Table 4). Similarly, under the SSP1-2.6 scenario, the surface from Colima and Tlaxcala is expected to be substantially reduced (100%), while Michoacán, Oaxaca, Veracruz, Chiapas, and Tamaulipas are likely to be considerably altered (about 22%); however, favourable climatic conditions may encourage environment improvement and expansion in other areas such as Jalisco, San Luis Potosí, Guerrero, Querétaro, Hidalgo, Estado de México, and Puebla (Table 4).

In the medium-term future with SSP5-8.5, climatic conditions are projected to decrease the surface area significantly in Jalisco state (96%). Additionally, the proportion of areas in Hidalgo and Morelos is also expected to diminish substantially, by around 60%. However, the species distribution is expected to increase in areas such as Tabasco, Querétaro, Michoacán, and Guerrero (Table 4). Similarly, in the future with SSP1-2.6, Guerrero and Estado de México are expected to reduce their distribution surface by around 80%, and Michoacán by 50%. Nevertheless, the environmental conditions are anticipated to significantly boost the areas in Tabasco, Querétaro, San Luis Potosí, Morelos, Puebla, Veracruz, and Oaxaca (Table 4).

On the other hand, the potential distribution of the species was found to be more associated with precipitation variables than temperature variables in the models. However, the second variable that contributed the most to the potential distribution ranged from 20.5 to 40.0% (Table 5). Analysis of temperature variables in the current scenario revealed that annual temperature oscillation is the main factor affecting the species distribution. However, in both medium-term and distant-future scenarios, this trend shifts to temperature seasonality. In the scenarios with a CO_2_ concentration of 2.6, temperature seasonality and annual temperature oscillation were identified as the principal variables for the distribution. Conversely, in scenarios with a CO_2_ concentration of 8.5, temperature seasonality and elevation played crucial roles. Notably, in scenarios with an 8.5 concentration, temperature seasonality remained the principal variable, along with elevation.

## 3. Discussion

### 3.1. The Impact of the Temperature Range on Seed Germination

The germination performance of *Leucaena diversifolia* over a temperature range was previously unknown before this research, with only one study recording the influence of temperature at 28 °C [26]. However, germination responses in a temperature range have been reported in other species from the same genus, e.g., *L. leucocephala* (Lam.) de Wit (20 to 70 °C) [12,27]. The higher germinations recorded at 25 and 30 °C (98 and 95%, respectively) in this study were higher than those reported by [26], which found 82% of germination at 26 days. Similarly, the germination percentage in *L. diversifolia* exceeded that reported by [27] for *L. leucocephala* at 40 °C (60%) but was similar to the percentages reported by [12,28] at lower temperatures (91% at 20 °C and over 95% at 24–28 °C). Based on the results, higher germination was observed at 25–35 °C, indicating that physiological processes may occur more efficiently within this range. The ability of *L. diversifolia* to germinate across a broad temperature range suggests that the species is well-adapted to different environmental conditions in the country.

Regarding the T_50_ values, they increased rapidly above 20 °C, while germination speed was slower in the 10–15 °C range. This result is consistent with studies in *L. leucocephala* which required 1.5–1.6 days to reach 50% of final germination across a range of 24–36 °C [28] and 3–4 days at 24 °C for *L. lanceolata* S. Watson [29]. These findings suggest that *L. diversifolia* could be germinated in warm environments under increased temperatures in medium-term and distant-future scenarios, potentially enabling rapid establishment in sites with mean monthly temperatures ranging from 20 to 30 °C. *L. diversifolia* is suitable for germination in a wide range of temperatures (5–39 °C), which is an important characteristic for facing climatic change, in particular for warmer climates. This highlights its ability to survive in sites with low and high temperatures and to cover an extended geographic distribution in the country. This adaptive feature is similar to that reported for some species from the Fabaceae family, such as *L. leucocephala*, *L. retusa*, *Vicia amoena* Fisch., *V. angustifolia* L., *V. sativa* L., and *V. unijuga* A. Braun [30,31,32]. The low *Tb* values (both models) in the target species may allow early spring sowings, thus ensuring quick establishment and reducing the risk of winter injuries [33]. These values are in line with those observed in previous studies for plants of the Fabaceae family such as *L. leucocephala* and *Onobrychis scrobiculata* Boiss. [28]. The *To* values estimated with both models were similar to those reported for the genus (25–30 °C [34]) but lower than those recorded by [35] for *L. leucocephala* (35 °C), while the *Tc* values were slightly lower than those recorded for *L. leucocephala* (46–47 °C [28]), and for two trees from the Fabaceae family: *Plathymenia foliolosa* Benth. and *Peltogyne confertiflora* (Mart. ex Hayne) Benth. (40 and 42 °C, respectively).

Overall, the thermal time provides a measure of the physiological time required for the completion of a process, such as germination, and it is expressed in the number of degree days (in °C). For the studied species, the thermal time at a sub-optimal temperature range pointed to a lower value compared to those recorded for three plants from the Fabaceae family: *Enterorlobium contortisiliquum* (Vell.) Morong, *Tachigali vulgaris* L.G. Silva and H.C. Lima, and *P. confertiflora*. This suggests that *L. diversifolia* is more sensitive to temperature and has a faster germination rate than the other species from the same family.

### 3.2. Differences in Germination Under Climate Change Scenarios

Climate change is clearly having an impact on the distribution of species in forest ecosystems. While climate is recognised as the main factor affecting diversity, other factors such as soil properties also play a crucial role in shaping community diversity and population sizes [36]. Despite the importance of this issue, there is a lack of studies that have used thermal modelling to describe the germination of species from the *Leucaena* genus. According to the models analysed in this study, the target species, *L. diversifolia*, is expected to maintain its distribution despite the increase in temperature, as it will not detrimentally affect the proportion of seeds that germinate. This finding is consistent with other studies conducted in warmer regions of Mexico and South America [22,23,37]. Based on the results, it is anticipated that *L. diversifolia* will attain its thermal sum (°Cd) accumulation more quickly. This finding is consistent with earlier studies on species such as *Swietenia macrophylla* King, *Cedrela odorata* L., *Polaskia chende* (Rol-.Goss.) A.C. Gibson and K.E. Horak, and *P. chichipe* (Rol.-Goss.) Backeb. [22,23,38].

### 3.3. Differences in the Potential Distribution

For *L. diversifolia,* both temperature and precipitation were identified as the main climatic variables affecting its distribution in the country, with temperature playing the most important role. These findings are consistent with a study by [39], which identified mean diurnal range (Bio2), isothermality (Bio3), temperature annual range (Bio7), and precipitation seasonality (Bio15) as key variables influencing species distribution. In future scenarios, changes in the populations of *L. diversifolia* are expected to reduce genetic diversity due to directional selection and rapid migration, potentially altering the resilience and functioning of ecosystems [40]. The high influence of temperature and precipitation variables observed in our study are consistent with findings for *L. leucocephala* reported by [41]. However, there were no reported common variables between *L. diversifolia* and *L. leucocephala*, despite being closely related species [42]. This discrepancy could be attributed to the fact that *L. diversifolia* inhabits zones with different environmental requirements compared to *L. leucocephala*. However, [43,44] found that two of the most significant factors influencing the global distribution of *L. leucocephala* were temperature seasonality (Bio4) and the minimum temperature of the coldest month (Bio6)., which aligns with our results for *L. diversifolia*. According to the MaxEnt model, temperature seasonality and elevation were among the variables with the highest contribution (19.7–31.6% and 13.9–15.3%, respectively) under different scenarios for *L. diversifolia*. These findings are consistent with those for *L. leucocephala*, where the same variables made significant contributions (24.6% and 13.3%, respectively) [45].

## 4. Materials and Methods

### 4.1. Plant Material

A portion of the State of Veracruz is included in the natural distribution range of Leucaena diversifolia in Mexico. Seeds were collected from the sub-deciduous tropical forest [46] in the Tlaltetela location (19°12′37.18″ N and 96°59′46.35″ W) at an elevation of 1517 m a.s.l. These seeds were considered as a representative sample of the trees from the region. The climate in this area is classified is Aw1″(w)(i)g [47], and the average temperature is 19 °C, with a mean annual precipitation of 2239 mm. Monthly temperatures range from 15.3 to 21.4 °C, while monthly precipitation ranges from 143 to 271 mm (source: http://es.climate-data.org/ accessed on 15 June 2022). A graphic depiction of these data is presented in Figure 7.

By manually cracking, the seeds were extracted from the pods and then washed. The seeds were stored at 5 °C in the Biotechnology and Prototype Research Unit’s Plant Physiology Laboratory (UBIPRO), Faculty of Higher Education Iztacala, National Autonomous University of Mexico (FES-I, UNAM), at Tlalnepantla, Mexico State. In order to increase the permeability of the seed coat, all seeds underwent scarification, which involved nicking the seeds with the edge of a nail clipper. This process was carried out to optimise germination.

### 4.2. Experiment Description and Experimental Design

Using 875 selected seeds, assays were carried out with 25 seeds placed in Petri dishes (5.5 × 1.5 cm) with agar medium (10 g L^−1^). Five replications of each temperature range (5–35 ± 2 °C, in 5-°C intervals) were conducted, yielding a total of 125 seeds for each temperature range. The Petri dishes were placed inside a germination chamber with an eight-hour photoperiod. The trial commenced on 9 March 2022 and followed a completely randomised design. Germination was recorded daily for a period of 35 days. When a seed radicle measured at least 2 mm in length, it was deemed to have germinated [48]. These recorded germination data were subsequently used to calculate the thermal times.

### 4.3. Data Analyses

The percentage of germination for each temperature range was calculated by averaging the germination data across the Petri dishes. After removing the empty seeds, the percentage of seeds that germinated was calculated using the following formula [49]:(1)$$G\left(\%\right)=\frac{n}{N}\times 100$$
where *G* represents the percentage of seeds that germinate, *n* denotes the number of seeds that germinate, and *N* represents the total number of viable seeds.

The number of days between the beginning of the imbibition and 50% of the total germination was recorded and subsequently used to calculate the mean germination time (T_50_). The accumulated germination data were fitted with a sigmoid curve, which made it possible to interpolate the median germination time [50]. The number of seeds that germinated each day, or the germination rate, was calculated using the formula below [51]:(2)$$GR=\frac{G_{1}}{N_{1}}+\frac{G_{2}}{N_{2}}+\cdots +\frac{G_{i}}{N_{i}}+\cdots +\frac{G_{n}}{N_{n}}=\sum_{n=1}^{n}\frac{G_{i}}{N_{i}}$$
where *GR* is the number of seeds that germinate per day, *G_i_* is the number of seeds that germinate in time *i*, and the time interval (in days) since the Petri dishes containing the seeds were put in the germination chamber is denoted by *Ni*.

For the temperature range, the proportion of germinated seeds, mean germination time (T_50_), and germination rate were evaluated using generalised linear models as those data were non-parametric. Statistical analysis was performed with the GENMOD procedure of the SAS^®^ version 9.3 program (SAS, Cary, NC, USA, 2010).

### 4.4. Cardinals Temperatures Calculation with Linear and Non-Linear Models

The reciprocal of the germination data at sub- and supra-optimal ranges vs. temperature was plotted to identify the various inflexion points for cardinal temperature determination, i.e., base temperature (*Tb*), optimal temperature (*To*), and ceiling temperature (*Tc*). The sub-optimal, optimal, and supra-optimal ranges showed statistically distinct changes in slope at 15 °C, 30 °C, and 35 °C, respectively. As a result, two distinct models with intersecting lines were run: one with two-segment regression lines (M1) and the other with simple regression lines (M2). The M1 model was computed using two two-segment linear regressions, one at the supra-optimal temperature range and the other at the sub-optimal temperature range; the intersection of these two lines indicated the *To* [52]. To determine the x-intercept of each regression line, or *Tb*, the first segment of the linear regression was used at the sub-optimal range, and the second segment was used at the supra-optimal range to determine the x-intercept, or *Tc*. [53]. The *Tb* and *Tc* were determined by averaging the x-intercept among fractions at the sub- and supra-optimal temperature range [54]. The first and second segments of the linear regressions in both temperature ranges were computed with the constraints X < X0 and X > X0, respectively, where X stands for any value of X and X0 stands for the value of the X coordinate at the intersection of the two segments. The inflexion points (X0) in the two-segment linear regressions were restricted to 15 °C and 35 °C.

The model with simple regression lines (M2) was constructed with linear regressions that covered the entire sub- and supra-optimal range. *Tb* and *Tc* were calculated from the average of the x-intercept in both ranges, while *To* was calculated from the intersection’s points of both regression lines. Mathematical modelling was based on the available literature [23,38,55,56,57].

The best regression models were chosen using the goodness of fit parameters, which included RMSE (Equation (3)), R^2^ (Equation (4)), and the intercept and slope of the regression of the predicted vs. observed germination rate.
(3)$$RMSE=\sqrt{{\left(\frac{1}{n}\right)\sum \left(Y_{obs}-Y_{pred}\right)}^{2}}$$
where Y_obs_ stands for observed value, *Y_pred_* for predicted value, and *n* is the number of samples [58].
(4)$$R^{2}=SSR/SST$$
where SSR stands for the sum of squares (SS) for each regression $\left(\sum_{i=1}^{n}\left(\hat{Y}-\bar{Y}\right)\right)$, and SST is the total SS $\left(\sum_{i=1}^{n}\left(Y_{i}-\bar{Y}\right)\right)$. *Y_i_* is used to represent the observed value, and *Y* is the associated estimated value. Better model estimation is indicated by low RMSE and R^2^ values close to 1.

The calibration of the models was performed by the iterative least squares method using TableCurve^®^ 2D (v5.01, Systat Software Inc., San Jose, CA, USA, www.sigmaplot.co.uk/products/tablecurve2d) (accessed on 20 May 2022) and GraphPad Prism^®^ v9.5.1 (San Diego, CA, USA; www.graphpad.com) (accessed on 24 May 2022).

### 4.5. Thermal Time (Sub- and Supra-Optimal)

The thermal time (*θ*) was calculated in the sub- and supra-optimal temperature range according to [22]. For every percentile, the supra-optimal time was determined by taking the inverse of the corresponding linear regression slope, and probits were calculated using the germination percentage data. Using the following equation, the sub-optimal thermal time was determined:(5)$$Probit\left(g\right)=K+\left[\theta_{1}/\sigma \right]$$
where *K* stands for the intercept constant when the germination progress is zero, and σ is the seed population response standard deviation at thermal time $\theta_{1}$. This equation was applied to the 50th percentile (*g*) to calculate the amount of time needed for 50% ($\theta_{1}$(50)) of the population to germinate.

The following formula was used to determine the cumulative temperature of the seeds needed for the germination process within the supra-optimal temperature range:(6)$$Probit\left(g\right)=Ks+\left(T+\theta_{2}/t\left(g\right)\right)/\sigma$$
where *Ks* stands for the intercept constant when the germination progress is zero; ($T+\theta_{2}/t\left(g\right)$) stands for the maximum temperature or *Tc*, and σ presents the deviation of the seed population response. This equation was applied to the 50th percentile (g) to calculate the amount of time needed for 50% ($\theta_{2}$(50)) of the population to germinate.

### 4.6. Seed Germination Under Climate Change Scenarios

Four average temperature layer projections from the Global Circulation Models were employed, according to [59]: the French model (CNRMCM5), American model (GFDL-CM3), English model (HADGEM2-ES), and German model (MPI-ESM-LR). According to [60], the models were created using the regional models of the Phase 5 Coupled Model Intercomparison Project (CMIP), developed by the Intergovernmental Panel on Climate Change (IPCC) (available in the Digital Climate Atlas for Mexico, http://atlasclimatico.unam.mx/cmip5/visualizador (accessed on 25 July 2022)). The models were projected for the medium-term future (2045–2069) and the distant future (2081–2100), using a monthly temporal and spatial resolution of 30″ × 30″ (roughly 926 × 926 m) and a Shared Socioeconomic Pathway (SSP1) value of 2.6 Watts/m^2^ (constant CO_2_ emissions), which was categorised as a conservative scenario.

The region was located on a map, and information on average temperatures for each Global Circulation Model (GCM), Shared Socioeconomic Pathway (SSP), and each future projection was collected. Average temperatures were projected for April, which is when seed dispersal starts, according to [61].

The time it will take for *Leucaena diversifolia* seeds to accumulate the thermal time needed for 50% of a seed bank in the understory to germinate was predicted for each scenario. In accordance with [62], the mean temperature method was used for the analysis, using the following equation [63]:(7)$$Thermal\;sum\;({}^{\circ}Cd)=\left(Env\;T_{m}-T_{b}\right)t_{m}$$
where *°Cd* stands for the accumulated degree days, *Tb* for the base germination temperature, *T_m_* for the average temperature for the month (m), and *t_m_* for the number of days in the month (m).

### 4.7. Potential Distribution Under Climate Change Scenarios

#### 4.7.1. Data Collection and Climatic Variables

The Global Biodiversity Information Facility (GBIF) platform of specimens deposited in herbaria worldwide provided the information needed to create a geo-referenced database of specimens collected in the country (https://www.gbif.org/, accessed on 10 September 2024). The database was then improved by removing references that were located in cities and incomplete data.

For the current distribution [64], nineteen climatic variables were used, each with a resolution of one km^2^ per pixel [65] and a spatial resolution of 0.3 min of arc, which were taken from the BioClim database.

Using information from the BioClim database, 19 climatic variables were used for the current distribution [64], with a spatial resolution of 0.3 min of arc and a resolution of 1 km^2^ per pixel [65]. Land use and vegetation vector format layers were obtained from the National Commission for the Knowledge and Use of Biodiversity [66].

#### 4.7.2. Current and Future Distribution

BioClim (v2.0) was used with data from 1970 to 2000 to determine the surface with the ideal climatic habitat for the current distribution. It was modelled using the MaxEnt 3.3.3 software [67]. This process is also known as environmental niche modelling [68,69] because it only considers climatic variables, and it could be defined as a climatic niche [70]. The relative contribution of each climatic variable to the model was determined using the Jackknife method, which was instrumented in MaxEnt. The software was run again to produce the maps using the climate factors that had a higher contribution to the model. Using the ArcMap 9.3^®^ tool, the spatial distribution of the ideal climatic habitat was determined by converting the number of pixels to km^2^. Finally, with a probability higher than 50%, maps of the species’ distribution in modern climates were produced.

BioClim v1.4 of was utilised to determine the surface with the ideal climatic habitat for future distribution. It was modelled using the MaxEnt v3.3.3 software [67]. The General Circulation Model (GCM) climate layers for the United States’ National Centre for Atmospheric Research (NCAR) (CCSM4) were downloaded, which, based on the IPCC’s regional models of the Coupled Models Intercomparison Project Phase 6, were projected for the years 2050 (average for 2041–2060) and 2090 (average for 2081–2099) as distant-future and medium-term future scenarios, respectively. The Jackknife method was applied with two Shared Socioeconomic Pathway (SSP) values of 2.6 and 8.5 Watts/m^2^ (lower and higher CO_2_ emissions) [59]. The software was run again to produce the maps using the climate factors that had a higher contribution to the model. Using the ArcMap 9.3^®^ tool, the spatial distribution of the ideal climatic habitat was determined by converting the number of pixels to km^2^. Finally, with a probability higher than 50%, maps of the species’ distribution in modern climates were produced.

#### 4.7.3. Validation of Distribution Models

In total, 30% of the data, which were randomly selected from the total number of locations where the species is found, were used in the test to validate the models. The average area under the receiver operating characteristics (ROC) curve (AUC) was used to evaluate the goodness of fit of the model predictions [71].

## 5. Conclusions

We confirmed that the temperature range 5–35 °C alters the germination, germination time, and germination rate of *L. diversifolia.* The responses of germination differed among temperatures. The observed differences in germination responses among temperatures underscore the importance of testing various thermal times to detect differences in accumulated days, particularly under climate change scenarios. These differences matter for conservation and restoration efforts in addition to seedling production. Understanding how temperature affects germination parameters is crucial for effectively managing and conserving plant populations, especially in the context of changing environmental conditions. The current area of *L. diversifolia* is projected to decrease in the future. The variables with the greatest contribution to the species’ distribution patterns were temperature, seasonality, and elevation. According to the model scenarios, states such as Colima, Tamaulipas, and Tlaxcala will be the most affected by climate change, experiencing a total reduction in the area with optimal conditions for the species. Furthermore, climate change models predict that the current distribution of the species in the study area may be reduced by up to 20%. For its conservation, it is recommended to collect seeds from populations in which ideal conditions for the species will no longer exist. Planting trials at different altitudes should also be carried out to facilitate assisted migration.

## Figures and Tables

**Figure 1 plants-13-02926-f001:**
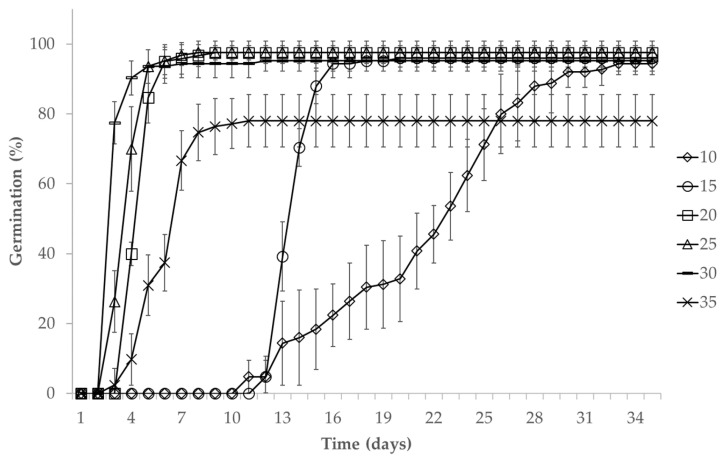
Cumulative germination curves of *Leucaena diversifolia* (Schltdl.) Benth. seeds at the temperature range 10–35 ± 2 °C. Germination at 5 °C was not observed.

**Figure 2 plants-13-02926-f002:**
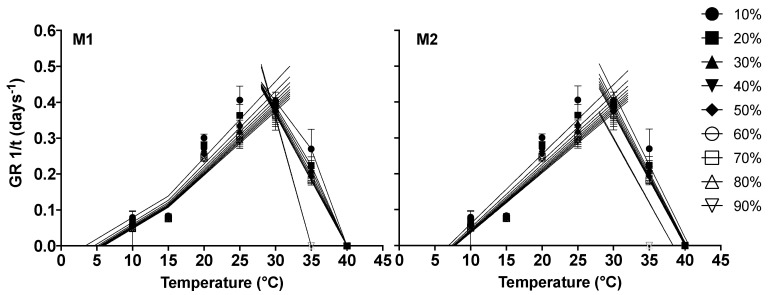
Correlation between 1/GR and germination temperature for 10–90% of seed population. Experimental data are represented by symbols, and predicted values are indicated by solid lines.

**Figure 3 plants-13-02926-f003:**
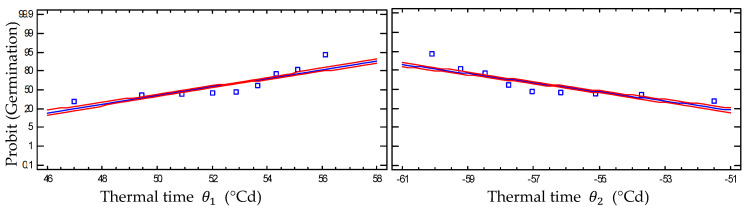
The Probit scale germination for the sub-optimal (left) $\theta_{1}$ and supra-optimal (right) $\theta_{2}$, temperature ranges as a function of thermal time. The red lines represent germination confidence intervals, while the estimated data are shown in the blue line. The points show the average of the experimental data.

**Figure 4 plants-13-02926-f004:**
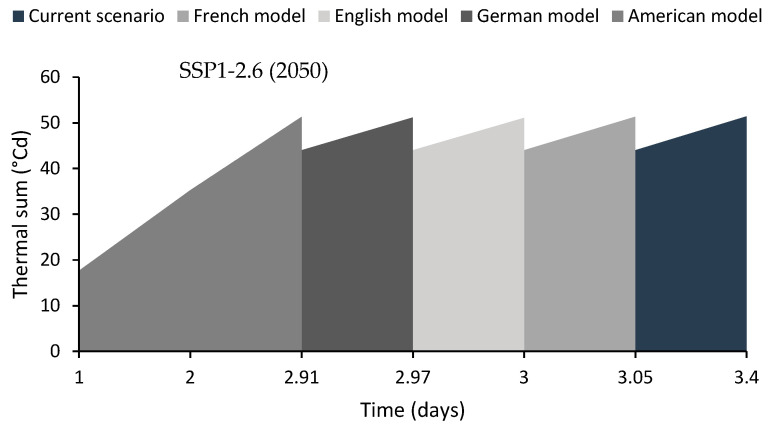
The length of time that seeds accumulate the thermal sum (°Cd) during April. The results were computed under a conservative scenario (SSP1-2.6 Watts/m^2^) and for an intermediate future (2050).

**Figure 5 plants-13-02926-f005:**
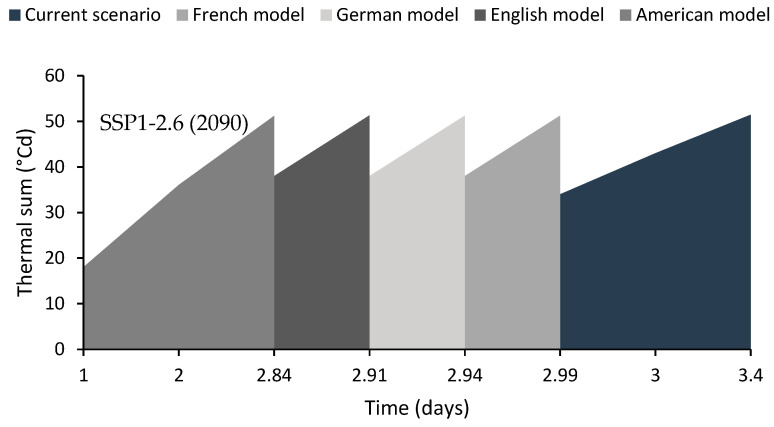
The length of time that seeds accumulate the thermal sum (°Cd) during April. The results were computed under a conservative scenario (SSP1-2.6 Watts/m^2^) and for a distant-future scenario (2090).

**Figure 6 plants-13-02926-f006:**
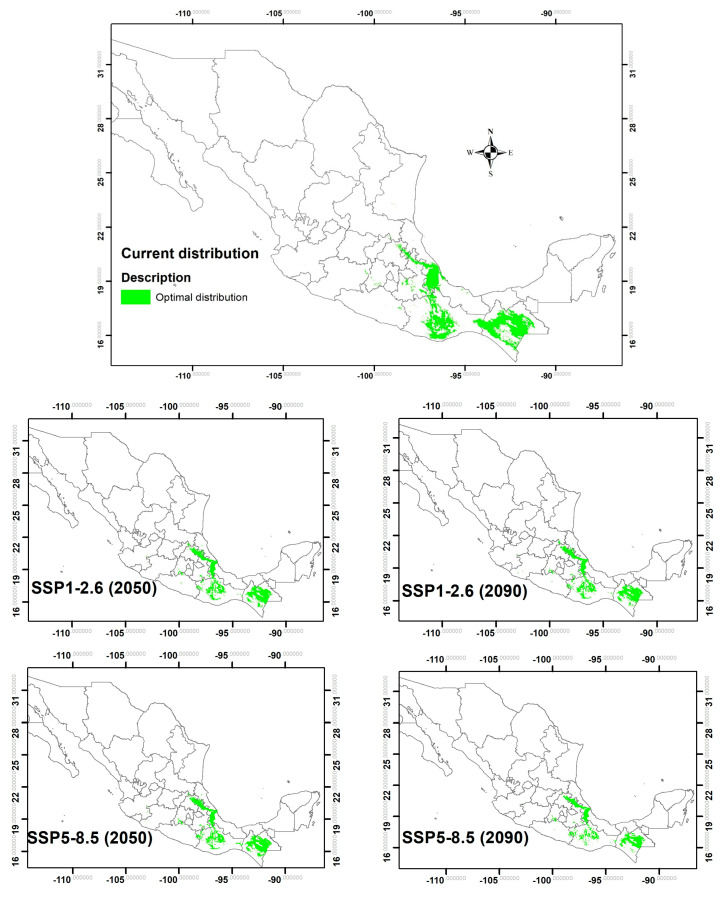
Potential distribution of *Leucaena diversifolia* (Schltdl.) Benth. in Mexico based on current and projected models for 2050 and 2090 using two Shared Socioeconomic Pathways of 2.6 and 8.5 Watts/m^2^.

**Figure 7 plants-13-02926-f007:**
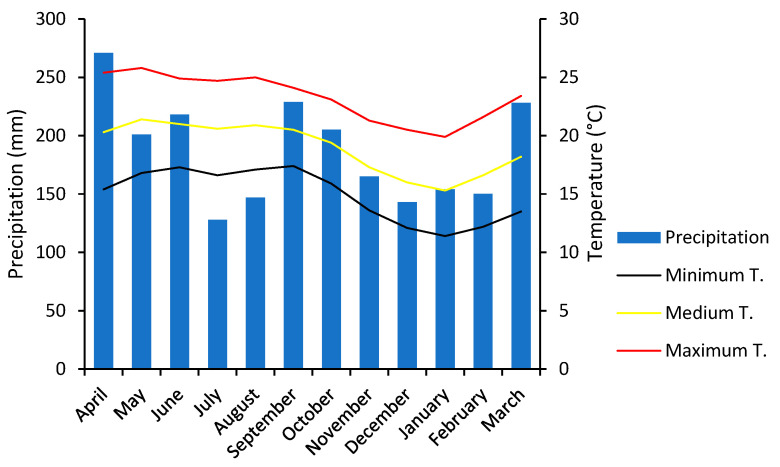
Climograph of monthly mean temperature (°C) and mean rainfall (mm) of Tlaltetela, Ver.

**Table 1 plants-13-02926-t001:** The Root Mean Square Error (RMSE), R^2^, and adjusted R^2^ for the two-segment linear regression model (M1) of *Leucaena diversifolia* (Schltdl.) Benth. seeds were estimated and are shown below.

Parameter	M1 (Two-Segment Linear Regressions)															
	10%		20%		30%		40%		50%		60%		70%		80%	
Tb (°C)	3.38		4.61		5.08		5.36		5.54		5.68		5.80		5.90	
Mean Tb (°C)	5.17 ± 0.83															
To (°C)	28.83		29.23		29.37		29.47		29.54		29.59		29.61		29.78	
Mean To (°C)	29.42 ± 0.29															
Tc (°C)	40.00		40.01		40.01		40.01		40.01		40.01		40.00		35.00	
Mean Tc (°C)	39.45 ± 1.67															
Range	Suboptimal	Supraoptimal	Suboptimal	Supraoptimal	Suboptimal	Supraoptimal	Suboptimal	Supraoptimal	Suboptimal	Supraoptimal	Suboptimal	Supraoptimal	Suboptimal	Supraoptimal	Suboptimal	Supraoptimal
RMSE	0.05	0.03	0.04	0.02	0.04	0.02	0.03	0.02	0.03	0.02	0.03	0.02	0.03	0.02	0.04	0.02
R^2^	0.87	0.97	0.91	0.99	0.92	0.99	0.93	0.99	0.94	0.99	0.94	0.99	0.94	0.99	0.92	0.99
Adjusted R^2^	0.86	0.96	0.90	0.99	0.92	0.99	0.93	0.99	0.93	0.99	0.93	0.99	0.94	0.99	0.91	0.99

**Table 2 plants-13-02926-t002:** The Root Mean Square Error (RMSE), R^2^, and adjusted R^2^ for the simple linear regressions model (M2) of *Leucaena diversifolia* (Schltdl.) Benth. seeds were estimated and are shown below.

	M2 (Simple Linear Regressions)							
Parameter	10%	20%	30%	40%	50%	60%	70%	80%
Tb (°C)	6.90	7.33	7.53	7.65	7.74	7.79	7.83	7.85
Mean Tb (°C)	7.60 ± 0.31							
To (°C)	29.62	29.60	29.68	29.70	29.71	29.72	29.74	28.61
Mean To (°C)	29.54 ± 0.38							
Tc (°C)	40.57	40.24	40.19	40.12	40.05	40.02	40.00	38.32
Mean Tc (°C)	39.76 ± 0.82							
RMSE	0.06	0.04	0.04	0.04	0.03	0.03	0.03	0.04
R^2^	0.86	0.90	0.92	0.93	0.93	0.93	0.94	0.91
Adjusted R^2^	0.86	0.90	0.91	0.92	0.93	0.93	0.93	0.91

**Table 3 plants-13-02926-t003:** Thermal time estimates based on the Probit analysis.

Parameter	Sub-Optimal	Supra-Optimal
R^2^	75.88	76.25
K	−9.068 ± 0.392	−10.343 ± 0.444
σ	0.176 ± 0.007	0.186 ± 0.007
$\theta \left(50\right)$ (°Cd)	51.34	55.57

The value represents the mean ± standard deviation.

**Table 4 plants-13-02926-t004:** Surfaces estimated by state for medium-term and distant-future outcomes with lower and higher CO_2_ emissions. The values were calculated using the Jackknife method according to the CCSM4 model.

State	Current (km^2^)	2050 (km^2^)		2090 (km^2^)	
		SSP1-2.6	SSP5-8.5	SSP1-2.6	SSP5-8.5
Chiapas	28,690.44	24,791.86	25,127.01	23,622.18	19,317.22
Oaxaca	21,014.39	16,533.54	17,108.19	15,894.36	12,079.69
Veracruz	11,381.13	9478.52	9610.30	9179.33	8717.22
Puebla	3368.95	4783.11	4944.93	4584.02	4247.42
Hidalgo	807.21	2907.00	2903.1	2678.28	2169.28
Estado de México	214.54	563.35	658.27	551.36	723.62
Guerrero	186.44	717.80	677.10	812.09	1031.55
Michoacán	152.36	20.73	90.84	96.30	8.58
Querétaro	48.55	215.87	113.44	188.37	47.62
San Luis Potosí	34.16	189.43	153.76	228.02	34.82
Tamaulipas	30.68	3.15	25.77	27.05	0.00
Tlaxcala	18.57	0.00	0.00	0.00	0.00
Jalisco	14.54	133.61	201.53	112.25	59.57
Tabasco	8.58	14.37	16.93	9.31	7.07
Colima	1.10	0.00	0.00	0.00	0.00
Campeche	0.00	11.32	6.16	6.16	0.00
Morelos	0.00	17.51	33.97	7.68	18.97
Nuevo León	0.00	8.30	2.10	1.05	0.00
Total	65,971.63	60,389.45	61,673.40	57,997.81	48,462.64

**Table 5 plants-13-02926-t005:** The percentage of the contribution of climatic variables in the models under different scenarios.

Current Scenario		Medium-Term Future (2050) 2.6 Watts/m^2^		Medium-Term Future (2050) 8.5 Watts/m^2^		Distant Future (2090) 2.6 Watts/m^2^		Distant Future (2090) 8.5 Watts/m^2^	
Variable ^†^	%	Variable	%	Variable	%	Variable	%	Variable	%
Bio7	28.2	Bio7	43.5	Bio7	43.3	Bio7	42.6	Bio7	44.2
Bio4	15.8	Alt	16.7	Alt	17.2	Alt	17.6	Alt	17.5
Alt	15.2	Bio15	7.5	Bio15	7.1	Bio15	8.4	Bio15	7.6
Bio15	8.6	Bio6	7.3	Bio6	6.5	Bio6	6.0	Bio13	5.8
Bio11	6.9	Bio13	5.1	Bio12	5.3	Bio13	5.6	Bio6	5.0
Bio12	5.7	Bio14	4.8	Bio14	5.1	Bio14	3.8	Bio14	3.8
Bio9	4.9	Bio4	4.3	Bio11	3.0	Bio4	2.5	Bio18	3.4
Bio14	3.5	Bio11	2.1	Bio4	2.4	Bio18	2.5	Bio11	2.6
Bio18	2.5	Bio12	1.7	Bio3	2.2	Bio11	2.2	Bio4	2.0
Bio13	1.9	Soil	1.7	Bio18	1.9	Bio3	2.0	Bio3	2.0
Soil	1.9	Bio19	1.5	Soil	1.9	Bio12	2.0	Soil	1.8
Bio2	1.8	Bio3	1.4	Bio8	1.8	Soil	1.8	Bio19	1.7
Bio3	1.3	Bio8	0.8	Bio19	1.5	Bio19	1.8	Bio8	0.9
Bio16	0.8	Bio18	0.6	Bio13	0.4	Bio16	0.4	Bio17	0.8
Bio8	0.7	Bio9	0.5	Bio2	0.2	Bio8	0.4	Bio9	0.5
Bio17	0.2	Bio16	0.3	Bio5	0.1	Bio2	0.2	Bio12	0.2
Bio19	0.1	Bio5	0.1	Bio16	0.1	Bio5	0.1	Bio2	0.1
				Bio9	0.1	Bio9	0.1	Bio16	0.1

^†^ Bio2: mean diurnal range, Bio3: isothermality, Bio4: temperature seasonality, Bio5: max temperature of warmest month, Bio6: minimum temperature of coldest month, Bio7: temperature annual range, Bio8: mean temperature of wettest quarter, Bio9: mean temperature of driest quarter, Bio11: mean temperature of coldest quarter, Bio12: annual precipitation, Bio13: precipitation of wettest month, Bio14: precipitation of driest month, Bio15: precipitation seasonality, Bio16: precipitation of wettest quarter, Bio17: precipitation of driest quarter, Bio18: precipitation of warmest quarter, Bio19: precipitation of coldest quarter, Alt: elevation.

## Data Availability

Data are contained within the article.

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
