# Peer review of "The Germination Performance After Dormancy Breaking of Leucaena diversifolia (Schltdl.) Benth. Seeds in a Thermal Gradient and Its Distribution Under Climate Change Scenarios"

_plants, 2024, doi:10.3390/plants13202926_

Round 1
Reviewer 1 Report
Comments and Suggestions for Authors
This manuscript evaluated the germination performance of Leucaena diversifolia under different temperatures using thermal times models. However, the primary issue with this manuscript is that the data and results presented in Figure 1 and Table 1 have already been published in the previous report (Cardinal temperatures of tree species used in agroforestry systems for coffee shade in Coatepec, Veracruz, Mexico). Figures 4 and 5 merely replace the SSP 2.6 model used in the previous study with the RCP 2.6 model, which does not provide a particularly meaningful result. Additionally, as per Section 4.6, the annotations in Figures 4 and 5 are incorrect. Similarly, Figures 2 and 3 do not offer much novelty compared to the earlier research. The remaining data in the manuscript are insufficient to warrant publication as an original research article.

Author Response
Comment 1
The primary issue with this manuscript is that the data and results presented in Figure 1 and Table 1 have already been published in the previous report (Cardinal temperatures of tree species used in agroforestry systems for coffee shade in Coatepec, Veracruz, Mexico).
Response
Dear reviewer, we appreciate your comment and recognize the sensitivity of this situation. The file referred to is a report on progress in knowledge that was available on the Pronatura website. It is important to mention that it is an internal report for the working group on the conclusion of the stated objectives, however, it does not constitute the publication of the article.
Additionally, I inform you that prior to sending the article, a review of similarities of the manuscript was carried out (attached report with 5% similarities) and the availability of this source was not detected, because the COMPILATIO software used searched regular sources of scientific journals.
To address this situation, the file has been removed from the Pronatura website.
We hope for your understanding in this situation of early dissemination of internal reports of the project.
Comment 2
The annotations in Figures 4 and 5 are incorrect.
We agree with the observation and appreciate your comment. In the revised version, the annotations in Figures 4 and 5 were changed from RCP 4.5 to SSP 1-2.6.

Reviewer 2 Report
Comments and Suggestions for Authors
Seed germination plays a very important role of plant growth. Although seed germination have been estimated for important species from Mexico,but there is no recorded information for Leucaena diversifolia. Thus, this experiment has certain application value.
some suggestions
1、The title of the experiment maybe be more suitable to change into the response of seeds to temperature after artificial breaking of dormancy.
lines 350“In order to increase the permeability of the seed coat, all seeds underwent scarification, which involved nicking the seeds with the edge of a nail clipper. This process was carried out to optimize germination.”
2、In the abstract and introduction, authors should further explain and elaborate on the scientific significance of this experiment.
Author Response
Comment 1
The title of the experiment maybe be more suitable to change into the response of seeds to temperature after artificial breaking of dormancy.
Response.
We appreciate the suggestion to modify the title. The change in the title was made following the reviewer's suggestion. The modified title better describes the content and importance of the manuscript.
Comment 2
In the abstract and introduction, authors should further explain and elaborate on the scientific significance of this experiment.
We agree with the suggestion to describe the scientific significance of the experiment better. Additional information was included in the abstract and introduction to explain the significance of the experiment, particularly on the importance of germination as the first stage for the recruitment of individuals in the communities of agroforestry systems.
Reviewer 3 Report
Comments and Suggestions for Authors
The work "Germination performance of seeds Leucaena diversifolia (Schltdl.) Benth. in a thermal gradient" is interesting and relevant, however the title of the manuscript should also reflect the climate change scenario.
Below are some points for attention by the authors:
1. Line 29-30: The sentence in the abstract section: "On the other hand, the supra-optimal and supra-optimal temperature values showed little differences: 51.34 and 55.57.", should be revised and corrected for thermal time (sub and supra); also the units for the values 51.34 and 55.57 should be added.
2. The following statement should be corrected: "Germination (radicle emergence) is the most critical stage of the plant life cycle [16] because it defines the efficient use of water and nutrients available for emergence [17].".
Although germination is a fundamental process for plant growth, it does not by itself define the efficiency of water use for emergence or the efficiency it will present throughout the life cycle of a plant. Please, correct the wording of the paragraph (lines 67 and 68).
3. Please reconcile or confirm the information in the figure caption (Figure 2) "Correlation between 1/GR and the germination temperature" and the y-axis title "GR 1/t (days-1)".
4. Table 3. Please confirm or correct the sign of the value "-55.57" of parameter ceta(50) in the supra-optimal column. It is recommended that the parameters where applicable have the units of measurement.
5. Figure 3. Translate the y-axis title into English and increase the font size.
Add the units to the x-axis variable.
6. Place bibliographical citations within the same pair of brackets (line 258).
7. The bracket in the quote [41] is missing. Line 320.
8. In Figure 7, review the minimum and maximum temperature data represented in the graph. The data legend appears to be wrong.
9. The conclusions section should include the study of climate change scenarios for L. diversifolia; and state what is expected regarding the distribution of the species in the study area.
Author Response
Comment
The work "Germination performance of seeds Leucaena diversifolia (Schltdl.) Benth. in a thermal gradient" is interesting and relevant, however the title of the manuscript should also reflect the climate change scenario.
Response
We agree with the suggestion to include the point on climate change in the title. The title in this revised version includes the climate change component.
Comment 1
Line 29-30: The sentence in the abstract section: "On the other hand, the supra-optimal and supra-optimal temperature values showed little differences: 51.34 and 55.57.", should be revised and corrected for thermal time (sub and supra); also the units for the values 51.34 and 55.57 should be added.
Response
We appreciate the suggestion to correct the temperature prefixes and the correction of the units in the text. The corrections were made in accordance with the reviewer's comments.
Comment 2
The following statement should be corrected: "Germination (radicle emergence) is the most critical stage of the plant life cycle [16] because it defines the efficient use of water and nutrients available for emergence [17].".
Although germination is a fundamental process for plant growth, it does not by itself define the efficiency of water use for emergence or the efficiency it will present throughout the life cycle of a plant. Please, correct the wording of the paragraph (lines 67 and 68).
Response
We agree with the reviewer's suggestion. Germination alone does not determine the efficient use of water and available nutrients throughout the plant's life cycle, so the correction was made to the text to indicate that it may affect the first stage of emergence and seedling development.
Comment 3
Please reconcile or confirm the information in the figure caption (Figure 2) "Correlation between 1/GR and the germination temperature" and the y-axis title "GR 1/t (days-1)".
Response
We appreciate the reviewer's comment. We confirm that the correlation described in the caption of Figure 2 (1/GR) is correct, which is the standard procedure for cardinal temperature models.
Comment 4
Table 3. Please confirm or correct the sign of the value "-55.57" of parameter ceta(50) in the supra-optimal column. It is recommended that the parameters where applicable have the units of measurement.
Response
We agree with the reviewer's suggestion. The sign of the theta value was changed to (55.57). In this sense, we commented that initially the negative sign had been included because the calculation occurs from a negative change in slope, but it is better interpreted with the positive sign.
Comment 5
Figure 3. Translate the y-axis title into English and increase the font size. Add the units to the x-axis variable.
Response
We thank the reviewer for this correction. The change to English language and description on the axes of Figure 3 was made.
Comment 6
Place bibliographical citations within the same pair of brackets (line 258).
Response
We thank the author for his comment. The change was made in the text as the journal's reference format.
Comment 7
The bracket in the quote [41] is missing. Line 320.
We thank the author for his comment. The change was made in the text as the journal's reference format.
Comment 8
In Figure 7, review the minimum and maximum temperature data represented in the graph. The data legend appears to be wrong.
Response
We agree with the correction indicated by the reviewer. The change was made to the Figure with the correct values.
Comment 9
The conclusions section should include the study of climate change scenarios for L. diversifolia; and state what is expected regarding the distribution of the species in the study area.
We thank the reviewer for his suggestion. Conclusions related to climate change were added, as well as some of the work's perspectives related to global warming.
Round 2
Reviewer 1 Report
Comments and Suggestions for Authors
The author has responded to the comments and revised the manuscript.
Author Response
Dear reviewer, we appreciate your comments on the manuscript.